# Single-atom Pt-I$_3$ sites on all-inorganic Cs$_2$SnI$_6$ perovskite for efficient photocatalytic hydrogen production

Peng Zhou[1,6], Hui Chen[1,2,3,6], Yuguang Chao[1,6], Qinghua Zhang[4], Weiyu Zhang[1], Fan Lv[1], Lin Gu[4], Qiang Zhao[2], Ning Wang [2,3], Jinshu Wang[2] & Shaojun Guo [1,5 ✉]

Organic-inorganic lead halide perovskites are a new class of semiconductor materials with great potential in photocatalytic hydrogen production, however, their development is greatly plagued by their low photocatalytic activity, instability of organic component and lead toxicity in particular. Herein, we report an anti-dissolution environmentally friendly Cs$_2$SnI$_6$ perovskite anchored with a new class of atomically dispersed Pt-I$_3$ species (PtSA/Cs$_2$SnI$_6$) for achieving the highly efficient photocatalytic hydrogen production in HI aqueous solution at room temperature. Particularly, we discover that Cs$_2$SnI$_6$ in PtSA/Cs$_2$SnI$_6$ has a greatly enhanced tolerance towards HI aqueous solution, which is very important for achieving excellent photocatalytic stability in perovskite-based HI splitting system. Remarkably, the PtSA/Cs$_2$SnI$_6$ catalyst shows a superb photocatalytic activity for hydrogen production with a record turnover frequency of 70.6 h$^{-1}$ *per* Pt, about 176.5 times greater than that of Pt nanoparticles supported Cs$_2$SnI$_6$ perovskite, along with superior cycling durability. Charge-carrier dynamics studies in combination with theory calculations reveal that the dramatically boosted photocatalytic performance on PtSA/Cs$_2$SnI$_6$ originates from both unique coordination structure and electronic property of Pt-I$_3$ sites, and strong metal-support interaction effect that can not only greatly promote the charge separation and transfer, but also substantially reduce the energy barrier for hydrogen production. This work opens a new way for stimulating more research on perovskite composite materials for efficient hydrogen production.

[1] School of Materials Science and Engineering, Peking University, Beijing, P. R. China. [2] School of Materials and Energy, University of Electronic Science and Technology of China, Chengdu, P. R. China. [3] State Key Laboratory of Marine Resource Utilization in South China Sea, Hainan University, Haikou, P. R. China. [4] Institute of Physics, Chinese Academy of Sciences, Beijing, P. R. China. [5] The Beijing Innovation Center for Engineering Science and Advanced Technology, Peking University, Beijing, P. R. China. [6] These authors contributed equally: Peng Zhou, Hui Chen, Yuguang Chao. ✉email: guosj@pku.edu.cn

Splitting hydroiodic acid (HI) has significant research value in the field of energy science and technology[1–5]. The traditional decomposition of HI at high temperature of 500 °C has been demonstrated to be an effective approach to produce hydrogen[2,5], however, it is unsustainable, dangerous, and not cost-effective. Solar-driven splitting of HI, as a promising low-cost technique, has recently attracted more research interest because it can achieve the co-production of zero-emission hydrogen ($H_2$) fuel and value-added chemicals ($I_2/I_3^-$) only under light condition at room temperature (RT)[6–8]. The development of advanced photocatalysts is very necessary for achieving the high efficiency of photocatalytic HI splitting, however, unfortunately, the reported photocatalysts cannot stably work in HI solution with strong acid property, which poses a high demand on new material selection and photocatalyst design. Recently, the organic–inorganic lead halide perovskites (OLHPs), such as $MAPbI_3$ (MA = $CH_3NH_3$), with the advantages of facile synthesis, low cost and superior optoelectronic characteristics[9–13], have been proved to be a promising photocatalyst for hydrogen production. Nevertheless, the unstable organic component in those $MAPbI_3$ photocatalytic materials easily suffers from the serious photo corrosion in HI solution[14–25], which severely limit the applications of OLHPs in the photocatalytic HI splitting into hydrogen[19,21]. Besides, the lead toxicity of $MAPbI_3$ also inhibits its practical application. In the seek for the lead-free perovskite materials, Sn-based perovskites have been demonstrated to have a narrower optical bandgap than that of the Pb-based perovskites[26–30], indicating their larger light absorption range. Especially, all-inorganic $Cs_2SnI_6$ perovskite is more preferred material system in view of its good stability, superior conductivity, and appropriate energy band levels[31–35].

Apart from above obstacles, the reported OLHPs-based photocatalysts still suffer from a very low photocatalytic activity caused by the serious photogenerated electron–hole recombination[15,17], restricting their further development. Decorating cocatalyst onto semiconductors to form the hetero-structured photocatalysts has been regarded as one of the simplest and most effective strategies for inhibiting the charge recombination, and hence improving the photocatalytic performance in the $H_2$ evolution reaction[36–40]. Considering the fact that halide perovskites possess a large number of defects caused by its inherent property of low temperature crystallization[41], they might be ideal scaffolds to anchor metal atoms and stabilize single atoms for achieving the greatly enhanced photocatalysis. However, developing new procedures for achieving highly efficient photocatalysts of halide perovskites anchored by a new class of metal single atoms for the hydrogen production, to the best of our knowledge, is still a great challenge in the field of photocatalysis.

Herein, we demonstrate the first example on making a new class of single-atom Pt–$I_3$ sites anchored on all-inorganic $Cs_2SnI_6$ perovskite (PtSA/$Cs_2SnI_6$) for efficient $H_2$ evolution photocatalysis from HI splitting at RT via a facile and cost-effective strategy. Turnover frequency (TOF) of as-prepared PtSA/$Cs_2SnI_6$ catalyst exhibits 176.5-fold enhancements compared with Pt nanoparticle anchored on $Cs_2SnI_6$ (PtNP/$Cs_2SnI_6$) catalyst, outperforming all of reported Pt-loaded halide perovskites photocatalysts, along with excellent catalytic stability. Combining charge-carrier dynamics studies with theory calculations reveals that both unique coordination structure and electronic property of Pt–$I_3$ sites and strong metal–support interaction (SMSI) effect are the main reasons in achieving the greatly enhanced photocatalytic performance in hydrogen production from HI aqueous solution.

## Results

### Energy band structure of $Cs_2SnI_6$ and its stability in aqueous HI solution system.
The synthetic procedure of PtSA/$Cs_2SnI_6$ is schematically illustrated in Fig. 1a. Briefly, the $Cs_2SnI_6$ was firstly synthesized by a one-pot hydrothermal treatment of cesium acetate and tin (II) acetate in presence of the excess hydriodic acid (HI) solution, followed by the impregnation of the platinum complex. Subsequently, the PtSA/$Cs_2SnI_6$ was obtained after activation at 160 °C for 1 h in $H_2$/Ar atmosphere. Field-emission scanning electron microscopy image of the as-prepared $Cs_2SnI_6$ demonstrates that the $Cs_2SnI_6$ mainly adopts the octahedral morphology (Fig. 1b). Powder X-ray diffraction (PXRD) pattern (Fig. 1c) of the obtained $Cs_2SnI_6$ can be well indexed to the cubic $Cs_2SnI_6$ phase (JCPDS card NO. 51-0466), confirming the successful synthesis of $Cs_2SnI_6$. Furthermore, the as-synthesized $Cs_2SnI_6$ with a high yield of 93% exhibits the high-temperature stability with a decomposition temperature up to 350 °C in air atmosphere (Supplementary Fig. 1). Furthermore, $Cs_2SnI_6$ is insoluble in aqueous HI solution at RT, confirmed by the fact that the solubility of $Cs_2SnI_6$ was zero at 25 °C, and slightly increased to $10.0 \times 10^{-6}$ mol $L^{-1}$ as the temperature was improved to 100 °C, still far lower than that of reported $MAPbI_3$ (0.645 mol $L^{-1}$ at 20 °C) (Fig. 1d)[25]. Moreover, there is no change in the color of aqueous HI solution with or without $Cs_2SnI_6$ powder, which further verifies the insolubility for $Cs_2SnI_6$ (inset of Fig. 1d).

Furthermore, the stability of $Cs_2SnI_6$ in aqueous HI solution system was further investigated via PXRD (Fig. 1e). We find that the as-prepared $Cs_2SnI_6$ powder exhibits the excellent stability in aqueous HI solution system (decompose when HI concentration is <1.0 M), better than that of reported $MAPbI_3$ (3.16 M)[25]. These results demonstrate that the $Cs_2SnI_6$-based aqueous HI splitting system is superior to the $MAPbI_3$-based one, suggesting the great potential in photocatalytic HI splitting. The ultraviolet–visible (UV–vis) absorption spectrum and X-ray photoelectron spectroscopy (XPS) valence spectrum were employed to determine the energy band structure of as-prepared $Cs_2SnI_6$. As shown in Fig. 1f, the $Cs_2SnI_6$ powder has a broad optical absorption range, and its bandgap energy ($E_g$) is 1.22 eV, by calculating from the absorbance data on the basis of the Kubelka–Munk equation, less than that of reported $MAPbI_3$ (1.53 eV) (Supplementary Fig. 2)[25]. The result indicates that $Cs_2SnI_6$ possesses a larger light absorption range and higher conductivity than $MAPbI_3$. The valence band (VB) position of the $Cs_2SnI_6$ is −5.46 eV with respect to the vacuum level (corresponding to 0.96 eV versus the normal hydrogen electrode (NHE)), as obtained from XPS VB spectroscopy (Supplementary Fig. 3). Accordingly, the corresponding conduction band (CB) position of $Cs_2SnI_6$, obtained by coupling the VB positions and $E_g$, is calculated to be −4.24 eV with respect to the vacuum level (corresponding to −0.26 eV versus NHE). It should be noted that the calculated values of CB and VB are an approximation since the complicated electrolyte–perovskite interface effects were not considered herein. The suitable energy band levels of $Cs_2SnI_6$ (inset of Fig. 1g) provides new opportunity for straddling the redox potentials of HI to split HI into $H_2$ and $I_3^-$.

### Structure characterization of PtSA/$Cs_2SnI_6$.
The aberration-corrected high-angle annular dark field-scanning transmission electron microscopy (HAADF-STEM) was performed to confirm the distribution and configuration of Pt single-atom in $Cs_2SnI_6$. No Pt clusters or large nanoparticles were observed in the low-magnification HAADF-STEM image (Fig. 2a), further confirmed by PXRD analysis (Supplementary Fig. 4), in which there are no characteristic peaks of crystalline Pt species. The atomic-resolution HAADF-STEM image (Fig. 2b) clearly depicts the individual dispersion of Pt atoms on the surface of as-prepared $Cs_2SnI_6$. The corresponding energy dispersive X-ray spectroscopy (EDS) mapping images confirm the uniform dispersion of Sn, I, and Pt atoms in PtSA/$Cs_2SnI_6$ (Fig. 2c), indicating that Pt atoms are uniformly distributed over $Cs_2SnI_6$. The Pt content on PtSA/

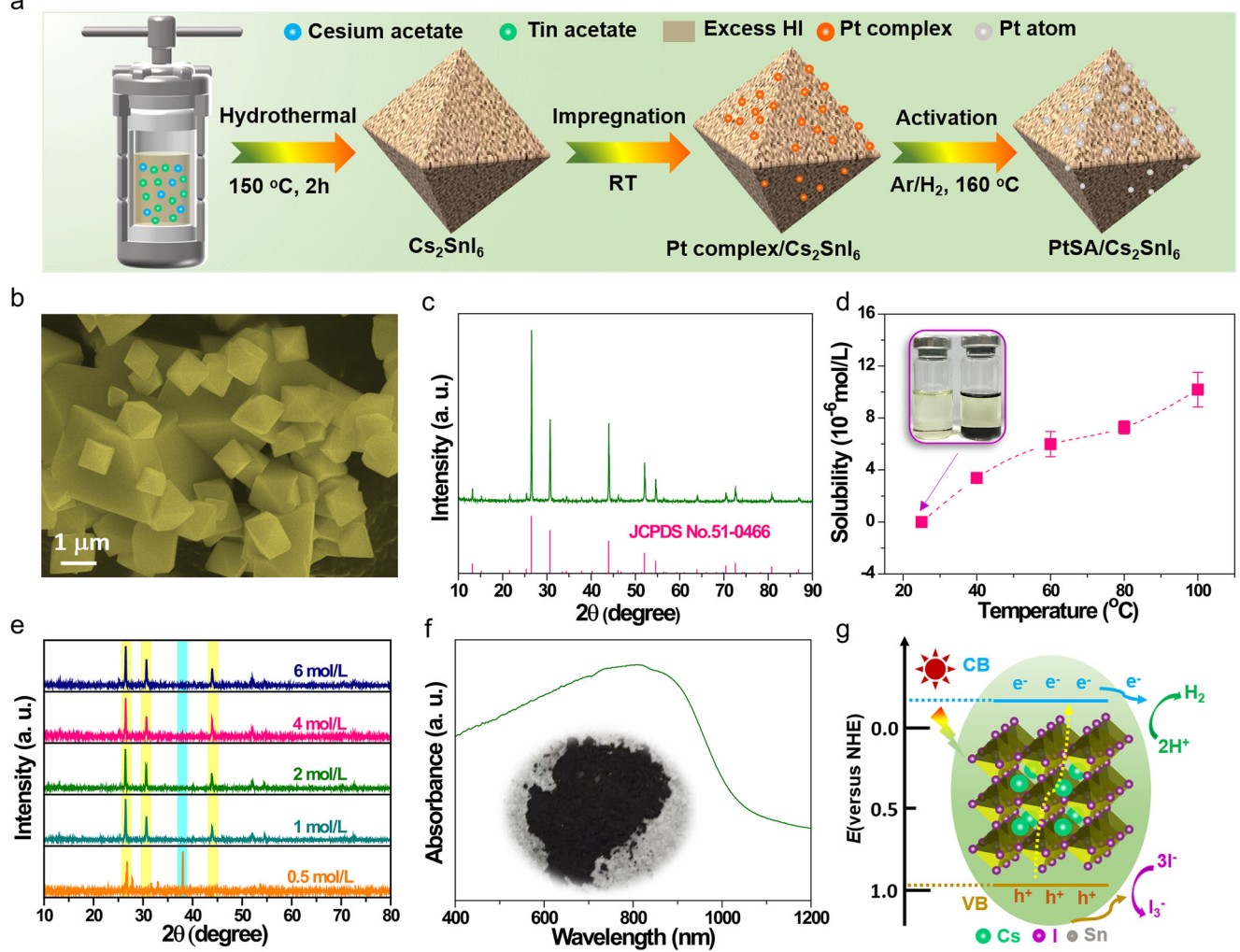

**Fig. 1 Energy band structure of Cs$_2$SnI$_6$ and its stability in aqueous HI solution system. a** Schematic diagram of preparation process for the PtSA/Cs$_2$SnI$_6$ catalyst. **b** SEM image and **c** PXRD pattern of Cs$_2$SnI$_6$. **d** Solubility of Cs$_2$SnI$_6$ in aqueous HI solution at different temperature. The inset shows the photograph of the 6 M HI aqueous solution with and without Cs$_2$SnI$_6$ powder at 25 °C. **e** PXRD patterns of precipitates for Cs$_2$SnI$_6$ powder in aqueous HI solution with various concentrations. **f** UV–visible absorption spectrum of Cs$_2$SnI$_6$ powder. The inset is the photograph of the Cs$_2$SnI$_6$. **g** Schematic energy band diagram of the Cs$_2$SnI$_6$, charge generation and charge transfer process over the Cs$_2$SnI$_6$ under visible-light irradiation.

Cs$_2$SnI$_6$ is determined to be 0.12 wt% by inductively coupled plasma-atomic emission spectrometry (ICP-AES).

The oxidation state and coordination environment of the Pt species in PtSA/Cs$_2$SnI$_6$ were further investigated by X-ray absorption fine structure (XAFS) spectroscopy. Figure 2d depicts the normalized Pt L$_3$-edge X-ray absorption near-edge structure (XANES) spectra of PtSA/Cs$_2$SnI$_6$, PtI$_2$ and Pt foil. The white-line intensity of Pt L$_3$-edge XANES of PtSA/Cs$_2$SnI$_6$ is higher than those of PtI$_2$ and Pt foil, indicating that the oxidation state of Pt in PtSA/Cs$_2$SnI$_6$ is more than +2. The Fourier-transformed (FT) k$^3$-weighted extended X-ray absorption fine structure (EXAFS) spectra (Fig. 2e) exhibits a prominent peak at ~2.49 Å in PtSA/Cs$_2$SnI$_6$, similar to that of the Pt–I bond in PtI$_2$, and shorter than that of the Pt–Pt bond in Pt foil, implying that the Pt atoms are anchored onto the surface of PtSA/Cs$_2$SnI$_6$ by the Pt–I bond. According to the EXAFS fitting results (Fig. 2f, Supplementary Figs. 5, 6, and Table 1), one Pt atom in PtSA/Cs$_2$SnI$_6$ is coordinated with approximately three I atoms (labeled as Pt–I$_3$). To further confirm the Pt single atoms in the PtSA/Cs$_2$SnI$_6$, the wavelet transform (WT) EXAFS of the PtSA/Cs$_2$SnI$_6$ and the reference systems (PtI$_2$ and Pt foil) was performed (Fig. 2g, h and

Supplementary Figs. 7, 8). The WT-EXAFS maximum is observed at 9.38 Å$^{-1}$ in Pt foil, assigned to the Pt–Pt bond. In contrast, the PtSA/Cs$_2$SnI$_6$ presents an intensity maximum value at 9.89 Å$^{-1}$, similar to that of Pt–I bond in PtI$_2$ (9.82 Å$^{-1}$), further indicating the atomic dispersion of Pt coordinated with I atoms on the surface of PtSA/Cs$_2$SnI$_6$.

To further confirm the chemical state of Pt and chemical environment of Cs, Sn, and I in the catalysts, XPS measurements were conducted. As shown in Fig. 2i, the binding energy of Pt 4$f$ in the high-resolution XPS spectrum of PtSA/Cs$_2$SnI$_6$ can be deconvoluted into two chemical states, assigned to Pt$^{2+}$ at 73.1 and 75.6 eV, and Pt$^{4+}$ at 75.1 and 77.5 eV, respectively, suggesting that Pt in the PtSA/Cs$_2$SnI$_6$ is Pt$^{\delta+}$ ($2 < \delta < 4$), in agreement with the aforementioned Pt L$_3$-edge XANES results. Noticeably, a positive shift in the I 3$d$ XPS spectra (Supplementary Fig. 9a) is observed upon Pt loading, implying the strong interaction between the Pt and I atoms in PtSA/Cs$_2$SnI$_6$. It means that those I species can serve as the anchor sites, and coordinate with the isolated Pt atoms. Moreover, the binding energy of Sn 3$d$ in PtSA/Cs$_2$SnI$_6$ has an obviously positive shift, whereas that of Cs 3$d$ was unchanged relative to that in pure Cs$_2$SnI$_6$

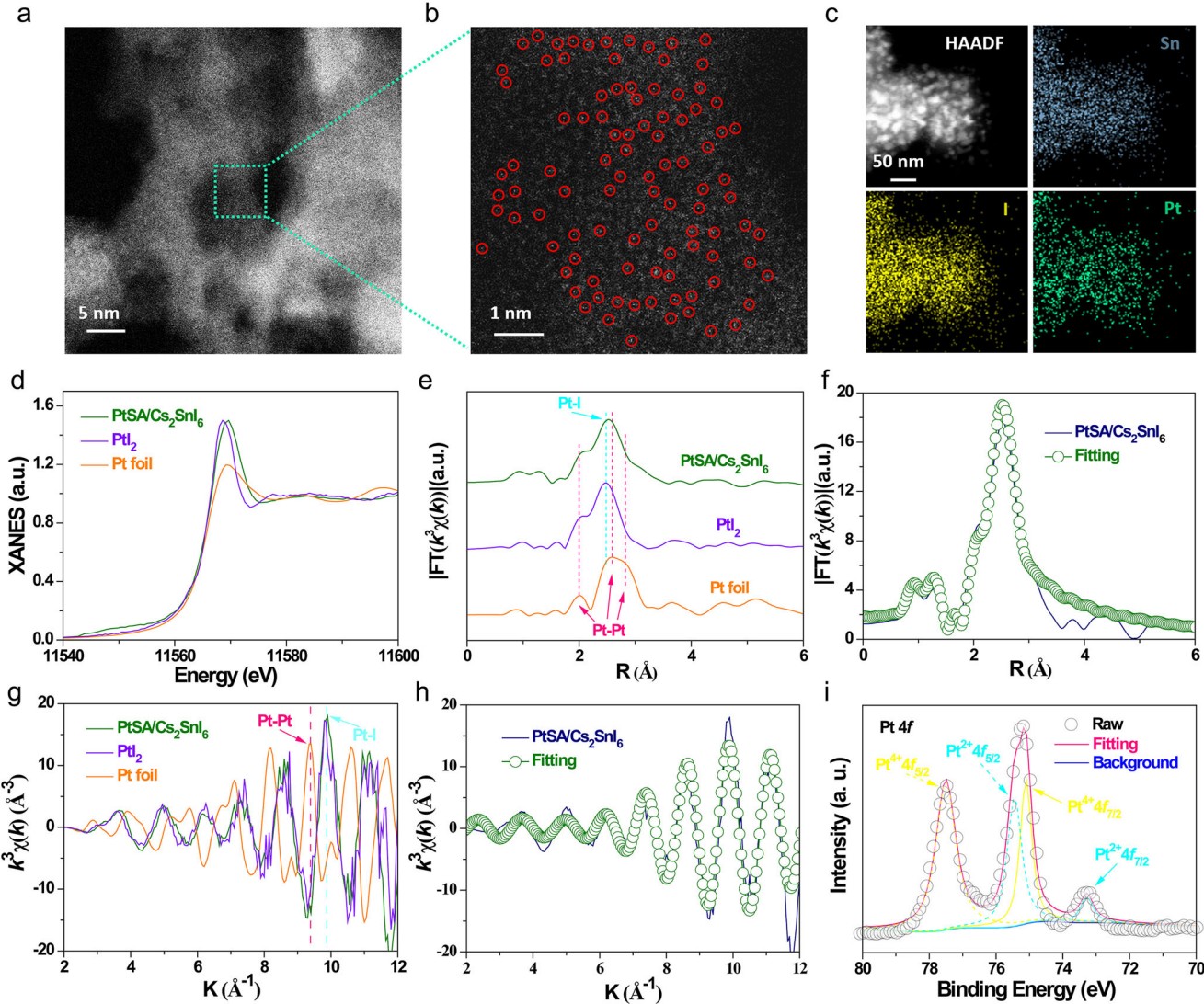

**Fig. 2 Structure characterization of PtSA/Cs$_2$SnI$_6$. a** Low-magnification and **b** high-magnification HAADF-STEM images, and **c** the corresponding STEM-EDS elemental mapping of PtSA/Cs$_2$SnI$_6$. **d** Pt L$_3$-edge XANES spectra and corresponding K$^3$-weighted FT spectra at **e** R space and **g** k-space of PtSA/Cs$_2$SnI$_6$, PbI$_2$ and Pt foil. XANES **f** R space and **h** k-space fitting curves of PtSA/Cs$_2$SnI$_6$. **i** High-resolution XPS Pt 4f spectrum of PtSA/Cs$_2$SnI$_6$.

(Supplementary Fig. 9b, c), implying that the electron transfer from Cs$_2$SnI$_6$ to PtSA depends on the Sn–I–Pt bonds rather than the Cs–I–Pt bonds.

**Superior photocatalytic activity and stability of PtSA/Cs$_2$SnI$_6$ catalyst**. The activities of photocatalytic H$_2$ evolution over PtSA/Cs$_2$SnI$_6$, PtNP/Cs$_2$SnI$_6$, and Cs$_2$SnI$_6$ catalysts were evaluated in aqueous HI solution system under visible-light ($\lambda \geq 420$ nm, 100 mW cm$^{-2}$) irradiation by a 300 W Xe lamp and a homemade double-layered Pyrex vessel. The results show that pristine Cs$_2$SnI$_6$ owns a poor photocatalytic performance with a H$_2$ production rate of 25 μmol h$^{-1}$ g$^{-1}$ (Fig. 3a and Supplementary Fig. 10). We find that the photocatalytic performance of Cs$_2$SnI$_6$ can be dramatically increased by anchoring Pt single atoms onto the surface of Cs$_2$SnI$_6$, reaching its maximum upon Pt loading up to 0.12 wt% (Fig. 3a and Supplementary Fig. 11). Further increasing or decreasing the loading content of Pt on Cs$_2$SnI$_6$ surface leads to the decreased photocatalytic activities. This is because the excessive Pt species can reduce the light absorption of Cs$_2$SnI$_6$ due to the shading effect whereas the insufficient Pt species cannot provide the rich H$_2$-releasing active sites.

Considering that the interfacial contact can affect the charge transfer from Cs$_2$SnI$_6$ to PtNP, we also prepared the PtNP on Cs$_2$SnI$_6$ (noted as PtNP$_{photo}$/Cs$_2$SnI$_6$) by the direct photo-deposition method, and studied its photocatalytic activity. The result shows that the rate of H$_2$ evolution (101 μmol g$^{-1}$ h$^{-1}$) over the optimal PtNP$_{photo}$/Cs$_2$SnI$_6$ sample is only slightly better than that of PtNP/Cs$_2$SnI$_6$ (Supplementary Fig. 12), still much lower than that of PtSA/Cs$_2$SnI$_6$. The 0.12 wt% PtSA/Cs$_2$SnI$_6$ shows the champion activity for the photocatalytic H$_2$ production with a H$_2$ production rate of 430 μmol h$^{-1}$ g$^{-1}$, 17.2 and 5.8 times higher than pure Cs$_2$SnI$_6$ and optimized 3.88wt% PtNP/Cs$_2$SnI$_6$ (Fig. 3a and Supplementary Figs. 13, 14), and also achieves a TOF of 70.6 h$^{-1}$ per Pt, 176.5 times higher than that of PtNP/Cs$_2$SnI$_6$ catalyst (0.4 h$^{-1}$) (Fig. 3b). Moreover, the TOF of PtSA/Cs$_2$SnI$_6$ catalyst is superior to all of the reported Pt-loaded halide perovskite photocatalysts (Fig. 3c and Supplementary Table 2).

Furthermore, the photocatalytic activity of PtSA/Cs$_2$SnI$_6$ can be stable without obvious decrease over four cycles and even at a successive 180-h tracking (Fig. 3d and Supplementary Fig. 15). After stability test, the Pt species are still atomically dispersed

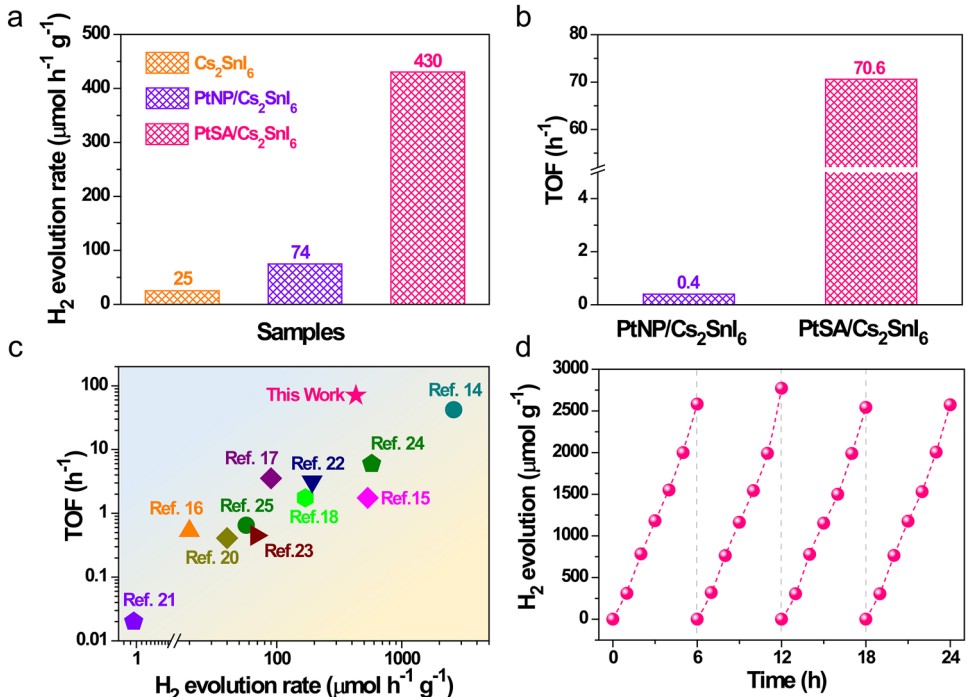

**Fig. 3 Superior photocatalytic activity and stability of PtSA/Cs₂SnI₆ catalyst. a** The rate of photocatalytic H₂ evolution over PtSA/Cs₂SnI₆, PtNP/Cs₂SnI₆, and Cs₂SnI₆ catalysts. **b** TOF of PtSA/Cs₂SnI₆ and PtNP/Cs₂SnI₆ catalysts. **c** TOF comparisons of PtSA/Cs₂SnI₆ catalyst and other reported Pt-loaded halide perovskite photocatalysts. **d** Cyclic stability of the PtSA/Cs₂SnI₆ catalyst.

on the surface of $Cs_2SnI_6$, confirmed by the diverse results of PXRD, HAADF-STEM, and STEM-EDS elemental mappings (Supplementary Fig. 16–18). In addition, the XPS spectrum of Pt 4*f* in $PtSA/Cs_2SnI_6$ after photocatalytic test reveals that the valence of Pt species has no obvious change (Supplementary Fig. 19), revealing the excellent chemical stability of PtSA on $Cs_2SnI_6$.

**Charge-carrier dynamics**. Photoluminescence (PL) technique was employed to evaluate the effect of Pt single atoms on the kinetics of charge carriers transfer and recombination over the $Cs_2SnI_6$. As depicted in Fig. 4a, the PL quenching of $PtSA/Cs_2SnI_6$ is more efficient than those of $PtNP/Cs_2SnI_6$ and $Cs_2SnI_6$, indicating the enhanced extraction and reduced recombination of charge carriers. To quantify the charge-carrier dynamic, TRPL spectroscopy was performed (Fig. 4b). The average decay lifetime of $PtSA/Cs_2SnI_6$ is ca. 61 ns (inset of Fig. 4b), smaller than those of $PtNP/Cs_2SnI_6$ (ca. 98 ns) and $Cs_2SnI_6$ (ca. 109 ns), further confirming that the atomically dispersed Pt species can enhance the separation of charge carriers. Besides, the photocurrent response spectra reveal that the $PtSA/Cs_2SnI_6$ exhibits a significantly enhanced photocurrent than those of $PtNP/Cs_2SnI_6$ and $Cs_2SnI_6$ (Fig. 4c), indicating its more efficient separation and transfer of photogenerated electron–hole pairs. Similarly, the linear-sweep voltammogram curves demonstrate that $PtSA/Cs_2SnI_6$ presents an overpotential of $-0.38$ V (versus RHE) at current density of 10 mA cm⁻², much lower than those of $PtNP/Cs_2SnI_6$ and $Cs_2SnI_6$ (Supplementary Fig. 20), implying its higher efficiency during the surface catalytic reaction. Furthermore, the $PtSA/Cs_2SnI_6$ displays the smallest semicircle in Nyquist plots obtained from the electrochemical impedance spectroscopy (Fig. 4d), also suggesting that the atomically dispersed Pt–I₃ configuration makes a great contribution to improving the efficiency of charge separation and transfer over $Cs_2SnI_6$.

**Theoretical calculations**. The photogenerated charge transfer on $PtNP/Cs_2SnI_6$ and $PtSA/Cs_2SnI_6$ was investigated by the DFT calculations. An extra electron was introduced to simulate the photogenerated electron in $PtNP/Cs_2SnI_6$ and $PtSA/Cs_2SnI_6$. The obtained charge density difference before and after photoexcitation reveals that the introduced photogenerated electrons tend to be distributed in the whole PtNP in $PtNP-Cs_2SnI_6$ (Fig. 5a), which undesirably decreases the electron density per Pt atom in PtNP. Instead, the electron in $PtSA/Cs_2SnI_6$ is only located between the PtSA and neighboring three I atoms (Fig. 5b), implying the high electron density on the Pt–I₃ site. This is attributed to the SMSI effect[42]. Thus, the electron density per Pt atom in PtNP is further lower than that of PtSA, which is considered to lead to the lower HER activity of PtNP. To exclude the background charge effect, one donor hydrogen atom was introduced to calculate the charge density difference[43]. The obtained results also showed the localized distribution of electrons in the Pt–I₃ region (Supplementary Fig. 21). This SMSI effect is beneficial to the photogenerated charge transfer between the photocatalyst and cocatalyst[42], also confirmed by the calculated PDOS of PtNP and PtSA on $Cs_2SnI_6$ (Fig. 5c). The most 5*d* states of PtSA are below the Fermi level, indicating its electron-rich state. The integrated PDOS areas of uncaptured Pt 5*d* states above the Fermi level in PtNP and PtSA are calculated to be 1.19 and 0.71, respectively, which quantifies the different electron-saturation levels of PtNP and PtSA. This implies the relative electron-deficient property of PtNP. This means that the PtSA species owns a stronger ability for capturing electrons from the $Cs_2SnI_6$, leading to the higher hydrogen production activity of $PtSA/Cs_2SnI_6$. Thus the different electronic properties of PtNP and PtSA on $Cs_2SnI_6$ lead to their different catalytic dynamics in the H₂ evolution process, in which the PtSA owns a remarkably lower energy barrier (0.11 eV) than PtNP (0.92 eV) (Fig. 5d). This well explains the observably higher photocatalytic H₂ production activity of $PtSA/Cs_2SnI_6$.

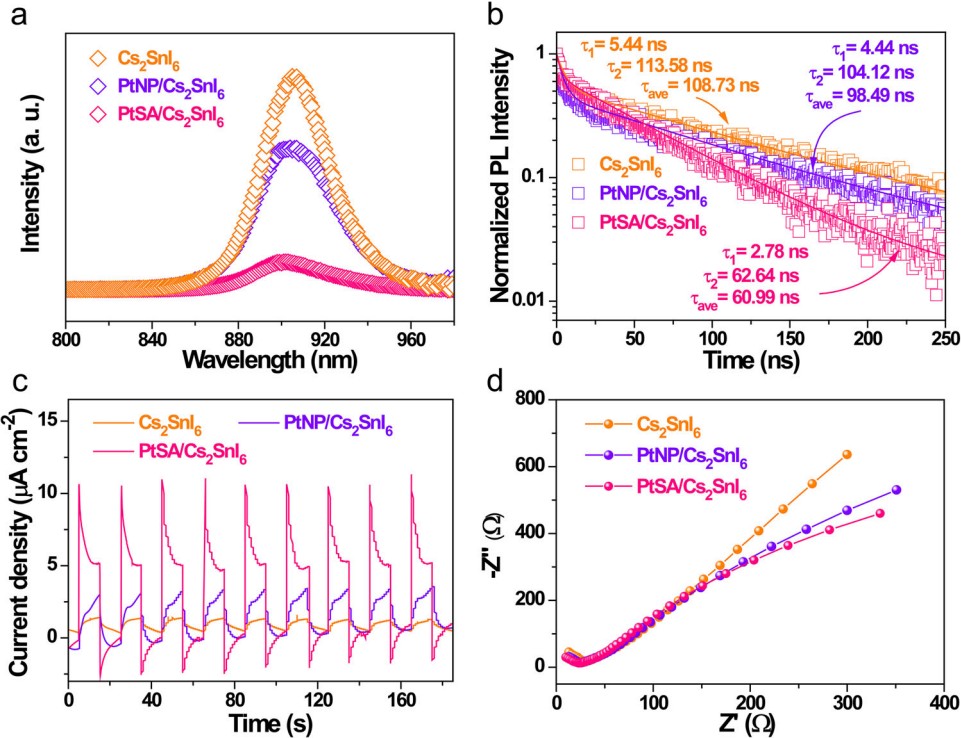

**Fig. 4 Charge-carrier dynamics. a** Steady-state PL spectra, **b** time-resolved transient PL decay, **c** photocurrent responses spectra, and **d** electrochemical impedance spectroscopy of $Cs_2SnI_6$, $PtNP/Cs_2SnI_6$, and $PtSA/Cs_2SnI_6$ catalysts.

## Discussion

In summary, we report a class of all-inorganic perovskite PtSA/ $Cs_2SnI_6$ single-atom photocatalyst for achieving highly efficient photocatalytic hydrogen production in aqueous HI solution. The HAADF-STEM, XAFS spectroscopy, and XPS spectroscopy results confirm the atomically dispersed Pt single atoms with the well-defined $Pt–I_3$ structure on $Cs_2SnI_6$. By combining charge-carrier dynamics studies and DFT calculations, we discover that the unique coordination structure and electronic property of $Pt–I_3$ species can contribute to the SMSI effect, boosting the photogenerated electron transfer from $Cs_2SnI_6$ to Pt single atoms, and simultaneously reducing the Gibbs free energy and accelerating the kinetics for the hydrogen production. Benefiting from these structural advantages, an outstanding TOF of 70.6 $h^{-1}$ per Pt was achieved over $PtSA/Cs_2SnI_6$ catalyst, 176.5-fold higher than that of $PtNP/Cs_2SnI_6$, setting a new TOF record reported for Pt-loaded halide perovskite photocatalysts, along with an excellent cycling stability. The achievements herein can significantly stimulate the exploitation of novel metal single atoms-perovskite hetero-structured photocatalysts systems and their further sustainable photocatalytic applications.

## Methods

**Synthesis of $PtSA/Cs_2SnI_6$.** As-prepared $Cs_2SnI_6$ powder (0.10 g) was dispersed in 50 mL of chloroform containing different amount of $Pt(acac)_2$. Then, the mixture was further ultrasonicated for 30 min and stirred for 12 h. Subsequently, the obtained precipitates were centrifugalized and washed for three times with isopropanol, and further dried in an oven 60 °C for 2 h. Finally, the $PtSA/Cs_2SnI_6$ was obtained by the treatment in a tube furnace at 160 °C for 1 h under a 5% $H_2/Ar$ atmosphere. The loading content of Pt on $PtSA/Cs_2SnI_6$ is determined by ICP-AES.

**XAFS characterization and data analysis.** The Pt $L_3$-edge XAFS spectra were performed at the 1W1B beamline of Beijing Synchrotron Radiation Facility (Beijing), operating at 2.5 GeV with a ring current of 250 mA. The X-ray beam was monochromatized by a Si (111) double crystal monochromator. The ATHENA module of the IFEFFIT software packages was used as the standard procedure to process the acquired EXAFS date. The $k^3$-weighted $\chi(k)$ data of Pt K-edge in the k-space (2.0–12 $Å^{-1}$) were FT to real (R) space by a handing windows (dk = 1.0 $Å^{-1}$)

to separate the EXAFS contribution from different coordination shells. The quantitative curve-fitting was performed by using the ARTEMIS module of IFEFFIT3 to obtain the elaborate structural parameters around Pt central atom in the as-synthesized catalysts. The functions of effective curved-wave backscattering amplitudes $F(k)$ and phase shifts $\Phi(k)$ were calculated by the ab initio code FEFF8.0. Based on the fitting of reference samples of metal Pt bulk and $PtI_2$ bulk, $S_0^2$ (amplitude reduction factor) was fixed to the best-fit value of 0.70. The interatomic distance (R) and the Debye–Waller factor ($\sigma^2$) were allowed to change during the fitting analysis. The coordination of Pt–I was distinguished from Pt–Pt according to the bond length difference.

**Photocatalytic $H_2$ evolution activity.** The photocatalytic $H_2$ evolution experiments in aqueous HI solution (containing 20 vol% $H_3PO_2$ as a stabilizer) were executed in a homemade double-layered Pyrex vessel. A 300 W Xe lamp with a visible-light illumination ($\lambda \geq 420$ nm, 100 mW $cm^{-2}$) was employed as a light source for the photocatalytic reaction. In a typical photocatalytic $H_2$ evolution procedure, 10 mg of as-synthesized photocatalyst was introduced into 10 mL aqueous HI solution containing 20 vol% $H_3PO_2$ with the constant stirring rate, and then degassed with Ar through the reactor for 30 min to completely remove the dissolved air before irradiation. The reaction temperature was preserved at 25 °C by a circulation cooling water. The amount of evolved $H_2$ was detected every hour in a 6 h test by gas chromatography (GC-7890B, Agilent, America, TCD, with MS-5 Å molecular sieve column) with Ar as the carrier gas. The recycling photocatalytic experiment for the stability test of as-prepared photocatalyst was carried out every 6 h as a cycle.

**Theoretical calculations.** The photocatalytic property of $PtSA/Cs_2SnI_6$ was investigated by the Vienna Ab initio Simulation Package. The PAW pseudo-potentials were used to describe the interaction between valence electrons and the ionic core. The energy profile for hydrogen production was calculated by the revised Perdew–Burke–Ernzerh functional of the generalized gradient approximation. The reported standard hydrogen electrode (SHE) model was used in the calculations of Gibbs free energy changes ($\Delta G$) in hydrogen adsorption[44]. In this model, the chemical potential ($\mu(H^+) + \mu(e^-)$) of a proton–electron pair was equal to half of the chemical potential ($\mu(H_2)$) of one gaseous hydrogen at $U = 0$ V versus SHE at pH = 0. The surface of $Cs_2SnI_6$ was simulated by its typical {111} facet, which was described by a 2 × 2 supercell with three Sn layers and six Cs–I layers (equal to three CsI–Sn–CsI layers). The PtSA was simulated by coordinating with three surface I atoms according to the above experimental results. The PtNP was described by a cluster containing 6 Pt atoms. The vacuum thickness was set to 16 Å. The larger supercells and higher vacuum thickness were tested, showing little effect on the calculation of hydrogen adsorption, as shown in Supplementary Table 3. A

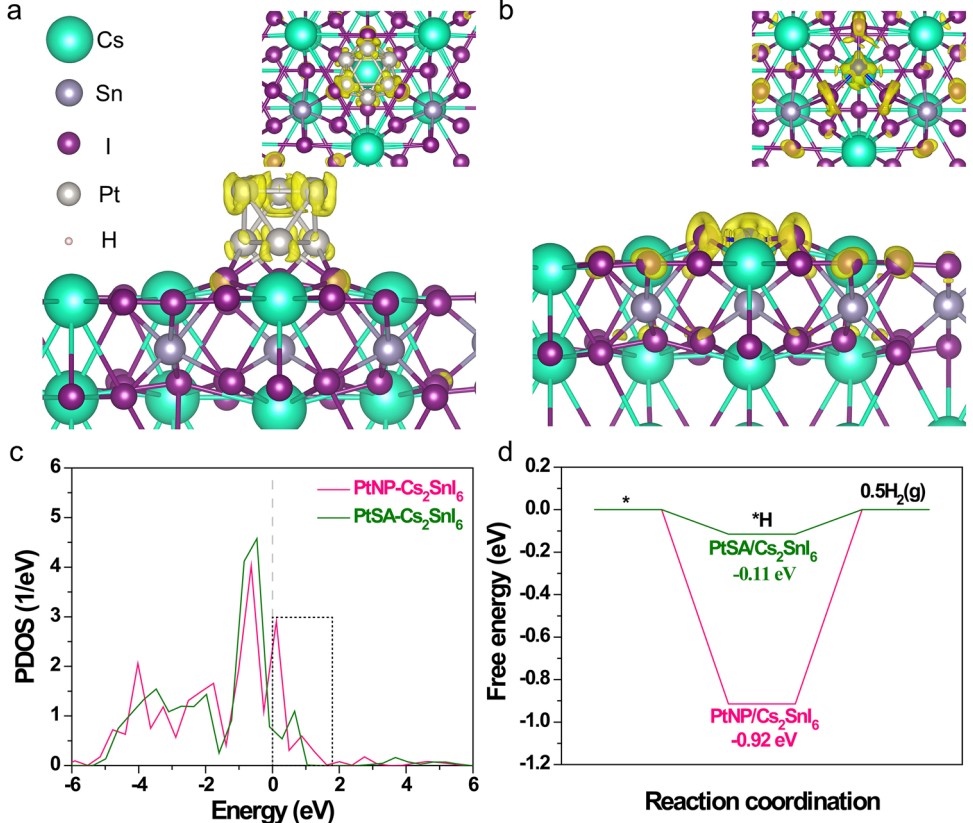

**Fig. 5 Charge density distribution and Gibbs energy calculations.** The charge density difference maps between before and after photoexcitation: **a** PtNP/ $Cs_2SnI_6$ and **b** PtSA/$Cs_2SnI_6$. The isosurface of charge density is 0.001e Å$^{-3}$. The insets stand for the top view. The yellow region represents the additional electron distribution. An excess electron was added into the models, which was used to describe the photogenerated electron. **c** The PDOS (5$d$ states) of PtNP/$Cs_2SnI_6$ and PtSA/$Cs_2SnI_6$. The dashed line stands for the Fermi level. **d** The calculated energy profile for hydrogen production on PtNP/$Cs_2SnI_6$ and PtSA/$Cs_2SnI_6$.

plane-wave basis with energy cutoff of 400 eV and an energy convergence threshold of $1.0 \times 10^{-5}$ eV were used to perform the geometry optimization at the gamma point. After geometry optimization, the projected density of states of PtNP/$Cs_2SnI_6$ and PtSA/$Cs_2SnI_6$ models is calculated with the energy convergence of $1.0 \times 10^{-5}$ eV and the Monkhorst–Pack k-point mesh of $2 \times 1 \times 1$. To track the transfer of photogenerated electron between the Pt and photocatalyst, an extra electron with a compensating uniform background charge was used to simulate the photo-generated electron[43]. The energy convergence of $1.0 \times 10^{-5}$ eV and the Monkhorst–Pack k-point mesh of $2 \times 1 \times 1$ were adopted to calculate the difference of charge density plots between PtNP/PtSA and $Cs_2SnI_6$. Besides, to exclude the background charge effect caused by the extra electron, the charge difference density maps between PtNP/PtSA and $Cs_2SnI_6$ was also calculated by inserting a donor hydrogen atom into the $Cs_2SnI_6$ bulk, which can provide one proton and one electron[43].

## Data availability

All experimental data within the article and its Supplementary Information are available from the corresponding author upon reasonable request.

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

## Acknowledgements
The authors are grateful for the financial supports of this work from the National Natural Science Fund for Distinguished Young Scholars (52025133), the Tencent Foundation through the XPLORER PRIZE, the Beijing Natural Science Foundation (JQ18005), the National Natural Science Foundation of China (22002003), the Fund of the State Key Laboratory of Solidification Processing in NWPU (SKLSP202004), and the China Post-doctoral Science Foundation (2019TQ0001 and 2020M670020). The authors also thank TianHe-2 of LvLiang Cloud Computing Center of China for theoretical calculations.

## Author contributions
S.G. conceived the project. P.Z., H.C., and Y.C. designed and performed the experiments. P.Z. carried out the DFT calculations and analysis. P.Z., H.C., Y.C., and F.L. conducted the XAFS characterization and corresponding data analysis. Q. Zhang and L.G. assisted with taking HAADF-STEM images. Q. Zhao, N.W., J.W., W.Z., and S.G. conducted data analyses and discussions. P.Z., H.C., and S.G. co-wrote the manuscript.

## Competing interests
The authors declare no competing interests.
