## [Peer Review File · Nature Communications]

REVIEWER COMMENTS

Reviewer #1 (Remarks to the Author):

In this manuscript, Chen and coworkers reported a high efficient photocatalyst for HI splitting based on Cs₂SnI₆ perovskite crystals with single atomic Pt-I₃ sites for accelerating the HER process. This optimized Pt SA/Cs₂SnI₆ catalyst delivered high photocatalytic activity of 430 $\mu\text{mol h}^{-1} \text{g}^{-1}$ and superior durability, which was comparable to MAPbI₃-based photocatalysts. More interesting, the turnover frequency of 70.6 h^{-1} per Pt was achieved in Pt SA/Cs₂SnI₆, which was about 176.5 times greater than that of Pt NPs supported Cs₂SnI₆ perovskite. It was demonstrated that the introduction of Pt-I₃ sites could promote the charge separation and transfer, and reduce the HER barrier. This work should also be inspirational for the design of perovskite-based materials with the combination of single atomic sites for other photocatalytic applications. Therefore, I would strongly recommend this work for publishing on Nat. Commun. after addressing the following concerns:

- (1) The solar-to-hydrogen conversion efficiency (STH) value of Pt SA/Cs₂SnI₆ should be provided.
- (2) The authors demonstrated a poor HER activity of Pt NPs/Cs₂SnI₆ by both its LSV curve (Supplementary Fig. 18) and theoretical calculation (Fig.), which was described as one of the key reasons for its low photocatalytic activity. However, I donot fully agree with this point, because it is well known that Pt-based nanomaterials are the best HER electrocatalysts. This inferior activity of Pt NPs/Cs₂SnI₆ may greatly relate to the possible sluggish electron transfer dynamics from CS₂SnI₆ to Pt NPs due to the mismatch between the conduction band of Cs₂SnI₆ and Fermi level of Pt NPs. The electrochemical signals of the LSV curve only reflected a generalized HER activity of the whole composite, and the actual HER activity of located Pt NPs may be suppressed due to its low content. For the theoretical calculation aspect, the provided model for Pt NPs was more like a Pt cluster, in which the underlying I- species could greatly affect the H* adsorption energy.
- (3) Furthermore, please add more details of the HER measurements in the experimental part, such as the used electrolyte and scanning rates. The description of the reference electrode (NHE) in the main text should be consistent with that in supplementary Fig. 18 (RHE).

Reviewer #2 (Remarks to the Author):

In this work, the authors reported the decoration of all-inorganic Cs₂SnI₆ perovskite with single-atom Pt-I₃ sites as cocatalyst (PtSA/Cs₂SnI₆) towards photocatalytic H₂ evolution from HI splitting reaction. The as-prepared PtSA/Cs₂SnI₆ catalyst showed a turnover frequency (TOF) 176.5 times higher than Cs₂SnI₆ decorated with Pt nanoparticle, along with excellent catalytic stability. Spectroscopy investigations together with theoretical calculations reveals that both unique coordination structure and electronic property of Pt-I₃ sites and strong metal-support interaction effect are the main reasons in achieving the greatly enhanced photocatalytic performance in hydrogen production from HI aqueous solution. The manuscript is well-organized and written and the results are interesting. I would suggest the acceptance of the work after the authors have addressed the following issues.

1. The PtNP/Cs₂SnI₆ was prepared by mixing pre-synthesized PtNp and Cs₂SnI₆ powder. In this case, the contact between PtNp and Cs₂SnI₆ powder should be poor, which will impair the interfacial charge transfer and therefore the photocatalytic activity. To make a fair comparison between PtNP/Cs₂SnI₆ and PtSA/Cs₂SnI₆, the PtNP/Cs₂SnI₆ sample could be prepared with photodeposition method to allow the direct growth of PtNP on Cs₂SnI₆. The photocatalytic activity of the as-prepared PtNP/Cs₂SnI₆ should be also tested.
2. The XPS spectra suggest that the valence state of Pt element in the sample is Pt²⁺ or Pt⁴⁺. However, as Pt²⁺ or Pt⁴⁺ can be easily reduced to Pt by photogenerated electrons during reaction, XPS analysis should be performed on the PtSA/Cs₂SnI₆ after phototcatalytic reactions to allow the

identification of the chemical state of Pt.

3. In Figure 4c, the conditions like the applied bias, the electrolyte (anything else besides 0.1 M TBAPF6 dichloromethane) for the photocurrent measurements should be provided. Moreover, as PtSA is used as the HER catalyst in this work to promote reduction reaction. I'm curious why positive photocurrents instead of negative photocurrents are used to testify the role of PtSA.

4. Some typos or errors in the manuscript should be corrected (Reference 23, 24 et al.).

Reviewer #3 (Remarks to the Author):

Hui Chen et al. reports on the photocatalytic properties of of a novel composite based on the Cs₂SnI₆ perovskite decorated with atomically dispersed Pt-I3 species. I have two main concerns about the manuscript. The first regards the possible appeal to a broad audience. In fact, the system used by the author, while possessing an improved stability in aqueous environment with respect to MAPbI₃ and showing interesting properties, still implies the use of highly concentrated HI solutions, thus severely limiting the range of applicability, which is restricted (e.g. no hydrogen production in more desirable conditions). Considering that perovskites which are actually water-stable have started to be synthesized (cf. ,<https://doi.org/10.1002/anie.202007584>), I fear that, without any interesting and motivated interpretation of novel phenomena, the manuscript in the current form, would be more suitable for a specialized journal.

Nevertheless, also in that case, there are some points that the authors need to address because the interpretation of their results appears to be shallow, in particular from the theory, and some results completely lack an explanation.

1) The computational details for the calculated results are frankly insufficient. The choice of adopted functional (which by the way does not include Van der Waals interactions that might play a role) is not motivated, convergence tests for the employed slab and the size of the vacuum layer are not reported.

2) Similarly, results are not clear. How do the authors simulate "before and after photexcitation" conditions in their calculations? Do calculation include extra electrons/holes? Then would a GGA treatment be sufficient to describe unpaired charges?

3) The authors reports charge density differences for the pristine material and PtSA but they do not include any comparison with PtNp. Furthermore, the localization of the charge in the figure is not clear. Are the author claiming that Cs₂SnI₆ feature a delocalized electronic state while, in presence of the additive, they observe a localized state in the band gap of the material?

4) I think that the claim that PDOS explains the difference between PtSA and PtNp is quite weak. I cannot see a dramatic difference among the two.

5) How do the authors calculate the energy diagrams in Fig. 5d?

6) The author state that "instability of the organic component" is at the root of the limited application of perovskites in photocatalysis. However, recent reports show that organic-inorganic perovskite can be water-stable.

7) The authors should comment on the trends observed for Pt loading? Why, after a maximum, the photocatalytic activity decreases?

Minor points:

(i) The authors should specify that the band alignment achieved via XPS is an approximation, since it does not include effects of the water-perovskite interface in the band alignment.

(ii) "The most 5d states of PtSA are below the Fermi level, indicating its strong electron-captured ability" this sentence is not clear and should be rephrased.

Overall, on the basis of my comments, I cannot suggest publication of the manuscript in the present form which need to be profoundly revised in order to explain the results

To Reviewer 1:

In this manuscript, Chen and coworkers reported a high efficient photocatalyst for HI splitting based on Cs₂SnI₆ perovskite crystals with single atomic Pt-I₃ sites for accelerating the HER process. This optimized Pt SA/Cs₂SnI₆ catalyst delivered high photocatalytic activity of 430 μmol h⁻¹ g⁻¹ and superior durability, which was comparable to MAPbI₃-based photocatalysts. More interesting, the turnover frequency of 70.6 h⁻¹ per Pt was achieved in Pt SA/Cs₂SnI₆, which was about 176.5 times greater than that of Pt NPs supported Cs₂SnI₆ perovskite. It was demonstrated that the introduction of Pt-I₃ sites could promote the charge separation and transfer, and reduce the HER barrier. This work should also be inspirational for the design of perovskite-based materials with the combination of single atomic sites for other photocatalytic applications. Therefore, I would strongly recommend this work for publishing on Nat. Commun. after addressing the following concerns:

R: Thanks for your great efforts in reviewing our manuscript. We specially appreciate your valuable comments and suggestions. We have performed all the experiments suggested by you, further addressed the comments point-by-point and made the corresponding changes accordingly in the revised manuscript.

Q1: The solar-to-hydrogen conversion efficiency (STH) value of Pt SA/ Cs₂SnI₆ should be provided.

R1: Thanks for your valuable comment. In general, the STH is mainly used in the uphill photocatalytic reaction, such as overall water splitting with the production of stoichiometric hydrogen and oxygen (H₂O → H₂ + 0.5O₂, ΔG = +237 kJ mol⁻¹). However, converting HI into hydrogen and by-product I₂ in our work is not an uphill reaction according to the chemical formula (2HI → H₂ + I₂, ΔG = -1.32 kJ mol⁻¹), which is mainly used for the hydrogen storage instead of solar-energy storage. In this regard, the STH may be not very necessary for the photocatalytic conversion of HI into hydrogen.

Q2: The authors demonstrated a poor HER activity of Pt NPs/Cs₂SnI₆ by both its LSV curve (Supplementary Fig. 18) and theoretical calculation (Fig.), which was described as one of the key reasons for its low photocatalytic activity. **(a)** However, I do not fully agree with this point, because it is well known that Pt-based nanomaterials are the best HER electrocatalysts. This inferior activity of Pt NPs/Cs₂SnI₆ may greatly relate to the possible sluggish electron transfer dynamics from Cs₂SnI₆ to Pt NPs due to the mismatch between the conduction band of Cs₂SnI₆ and Fermi level of Pt NPs. **(b)** The electrochemical signals of the LSV curve only reflected a generalized HER activity of the whole composite, and the actual HER activity of located Pt NPs may be suppressed due to its low content. **(c)** For the theoretical calculation aspect, the provided model for Pt NPs was more like a Pt cluster, in which the underlying I⁻ species could greatly affect the H* adsorption energy.

R2: Thanks for your valuable comments.

(a) Pt is considered as the best HER electrocatalysts. However, the reaction activity of Pt cocatalyst in photocatalytic reaction is also strongly determined by the interaction between photocatalyst surface and cocatalyst. The excellent activity of PtSA in our work is attributed to its special coordination structure since the PtSA is completely coordinated with the photocatalyst surface *via* three Pt-I bonds. This contributes to a strong interface between PtSA and Cs₂SnI₆, which is beneficial to the direct charge transfer from Cs₂SnI₆ to PtSA. However, at the PtNP-Cs₂SnI₆ interface, only the interfacial Pt atoms can be coordinated with Cs₂SnI₆ surface. Unfortunately, those surface catalytic Pt sites cannot be directly coordinated with the Cs₂SnI₆ surface, which undesirably increase the charge transfer distance or barrier. Thus the PtSA and PtNP with different cocatalyst-photocatalyst interactions lead to their different electronic and catalytic properties. This has been well explained in the **Line 17-24 of Page 8**.

(b) The optimization on PtSA and PtNP have been done in **Supplementary Fig. 11** and **Supplementary Fig. 14**. It is obvious that the activity of optimized PtNP-Cs₂SnI₆ with 3.88wt% Pt is still lower than that of PtSA-Cs₂SnI₆ with only 0.12wt% Pt. Hence, the difference between the activities of PtSA and PtNP on

Cs₂SnI₆ cannot be simply attributed to their different Pt contents. Instead, the coordination structure of Pt species plays a significant role in the photocatalytic reaction. The detailed discussion has been added in the **Line 3-8 of Page 7**.

(c) Considering the size effect, a larger Pt nanoparticle consisting of 31 Pt atoms was used in the model, which also show a more negative hydrogen adsorption energy (-0.85 eV) than that of PtSA, as shown in **Supplementary Table 3**. This suggests that the PtSA on Cs₂SnI₆ owns a higher activity than Pt cluster or nanoparticle.

Q3: Furthermore, please add more details of the HER measurements in the experimental part, such as the used electrolyte and scanning rates. The description of the reference electrode (NHE) in the main text should be consistent with that in supplementary Fig. 18 (RHE).

R3: Thanks for your valuable comments. The HER measurements were performed in the HI and H₃PO₂ mixed solution (57 wt% HI 16 mL + 50 wt% H₃PO₂ 4 mL) at a scan rate of 50 mV s⁻¹. We have added these details in the revised supporting information (Please see the **Line 11-14 of Page 3 of the revised supporting information**). The overpotential is calculated based on the reversible hydrogen electrode (RHE), we have revised the calculation description in the revised manuscript (Please see **Line 6-11 of Page 8**).

To Reviewer 2:

In this work, the authors reported the decoration of all-inorganic Cs₂SnI₆ perovskite with single-atom Pt-I₃ sites as cocatalyst (PtSA/Cs₂SnI₆) towards photocatalytic H₂ evolution from HI splitting reaction. The as-prepared PtSA/Cs₂SnI₆ catalyst showed a turnover frequency (TOF) 176.5 times higher than Cs₂SnI₆ decorated with Pt nanoparticle, along with excellent catalytic stability. Spectroscopy investigations together with theoretical calculations reveals that both unique coordination structure and electronic property of Pt-I₃ sites and strong metal-support interaction effect are the main reasons in achieving the greatly enhanced photocatalytic performance in hydrogen production from HI aqueous solution. The manuscript is well-organized and written and the results are interesting. I would suggest the acceptance of the work after the authors have addressed the following issues.

R: Thanks for your great efforts in reviewing our manuscript. We specially appreciate your valuable comments and suggestions. We have performed all the experiments suggested by you, further addressed the comments point-by-point and made the corresponding changes accordingly in the revised manuscript.

Q1: The PtNP/Cs₂SnI₆ was prepared by mixing pre-synthesized PtNP and Cs₂SnI₆ powder. In this case, the contact between PtNP and Cs₂SnI₆ powder should be poor, which will impair the interfacial charge transfer and therefore the photocatalytic activity. To make a fair comparison between PtNP/Cs₂SnI₆ and PtSA/Cs₂SnI₆, the PtNP/Cs₂SnI₆ sample could be prepared with photodeposition method to allow the direct growth of PtNP on Cs₂SnI₆. The photocatalytic activity of the as-prepared PtNP/Cs₂SnI₆ should be also tested.

R1: Thanks for your valuable comment. Indeed, the interfacial contact can affect the charge transfer from Cs₂SnI₆ to PtNP. According to your nice suggestion, we have supplemented the photocatalytic activity of PtNP_{photo}/Cs₂SnI₆, prepared by the photo-deposition method. The rate of H₂ evolution over the optimal PtNP_{photo}/Cs₂SnI₆ sample is 101 μmol g⁻¹ h⁻¹, which is better than that of PtNP/Cs₂SnI₆, but still much lower than that of PtSA/Cs₂SnI₆ sample. This has been added into the revised manuscript (Please see **Line 8-13 of Page 7 of revised manuscript, Line 19-22 of Page 2 of revised supporting information and Supplementary Fig. 12**).

Supplementary Fig. 12. Photocatalytic H₂ evolution rate of PtNP_{photo}/Cs₂SnI₆ depending on the loading amount of Pt.

Q2: The XPS spectra suggest that the valence state of Pt element in the sample is Pt²⁺ or Pt⁴⁺. However, as Pt²⁺ or Pt⁴⁺ can be easily reduced to Pt by photogenerated electrons during reaction, XPS analysis should be performed on the PtSA/Cs₂SnI₆ after photocatalytic reactions to allow the identification of the chemical state of Pt.

R2: Thanks for your valuable comment. The positive-valence Pt species can accept the electron, and be reduced in the photocatalytic reaction. However, this is a dynamic process. The photogenerated electrons aggregated in Pt would be used to reduce protons into hydrogen. After photocatalytic reaction, the Pt species return to its initial chemical state. This has been demonstrated in the supplemented XPS test, in which the Pt 4f high resolution XPS spectrum of the sample after photocatalytic reaction is similar to that of the sample before photocatalytic reaction (Please see Line 23-25 of Page 7 and Supplementary Fig. 19).

Supplementary Fig. 19. The XPS spectra of Pt 4f in PtSA/Cs₂SnI₆ before and after photocatalytic reaction.

Q3: (a) In Figure 4c, the conditions like the applied bias, the electrolyte (anything else besides 0.1 M TBAPF₆ dichloromethane) for the photocurrent measurements should be provided. (b) Moreover, as PtSA is used as the HER catalyst in this work to promote reduction reaction. I'm curious why positive photocurrents instead of negative photocurrents are used to testify the role of PtSA.

R3: Thanks for your valuable comments.

(a) The photocurrent responses were measured by utilizing 300W Xe lamp with an ultraviolet cut-off filter ($\lambda \geq 420$ nm) as the light source at 0 V vs. Ag/AgCl reference, and 0.1 M TBAPF₆ dichlormethane solution as electrolyte. We have added these details in the revised supporting information (Please see the **Line 11-14 of Page 3** of the revised supporting information).

(b) Cs₂SnI₆ is n-type semiconductor, which produces the positive photocurrent under present weak bias condition. The photocurrent test is mainly used to investigate the charge separation/transfer ability of photocatalyst instead of surface catalytic redox reaction. In our manuscript, the absolute value of current variation before and after illumination represents the carrier separation/transfer efficiency, which is influenced by the electronic properties of photocatalytic bulk and surface. The results show that the PtSA modification can significantly promote the charge separation/transfer efficiency between photocatalyst bulk and surface. Moreover, the surface catalytic redox reaction is mainly evaluated by the LSV test with negative current density as shown in **Supplementary Fig. 20**, which demonstrates the promotion effect of PtSA. This discussion has been added into the revised manuscript (please see **Line 6-11 of Page 8**).

Q4: Some typos or errors in the manuscript should be corrected (Reference 23, 24 et al.).

R4: Thanks for your valuable comments. We have carefully checked and revised all typos and errors in the reference section in the revised manuscript.

To Reviewer 3:

Hui Chen *et al.* reports on the photocatalytic properties of a novel composite based on the Cs₂SnI₆ perovskite decorated with atomically dispersed Pt-I₃ species. I have two main concerns about the manuscript. The first regards the possible appeal to a broad audience. In fact, the system used by the author, while possessing an improved stability in aqueous environment with respect to MAPbI₃ and showing interesting properties, still implies the use of highly concentrated HI solutions, thus severely limiting the range of applicability, which is restricted (*e.g.* no hydrogen production in more desirable conditions). Considering that perovskites which are actually water-stable have started to be synthesized (cf. <https://doi.org/10.1002/anie.202007584>), I fear that, without any interesting and motivated interpretation of novel phenomena, the manuscript in the current form, would be more suitable for a specialized journal. Nevertheless, also in that case, there are some points that the authors need to address because the interpretation of their results appears to be shallow, in particular from the theory, and some results completely lack an explanation.

R: Thanks for your great efforts in reviewing our manuscript. We specially appreciate your valuable comments and suggestions. We believe that the present work would be of great and instant importance to the following research fields:

(1) Materials Science and Nanoscience. Designing the high-performance metal single atom photocatalytic catalysts is gaining dramatic attention. However, the common nitrogen or oxygen-coordinated metal single atom structure in the reported single atom catalysts still make them show the limited catalytic activity due to the high electronegativity of those ligand atoms. Herein, we firstly synthesize and demonstrate a new type of atomically dispersed iodine-coordinated Pt single atom structure (Pt-I₃) with electron-rich feature, which is very promising for solar hydrogen photocatalytic production. Therefore, this work may raise research enthusiasm in the communities who are interested in single atom catalysis, materials science and nanoscience, *etc.*

(2) Hydrogen energy conversion. Though there are few reports on water-stable perovskite photocatalysts pointed by the reviewer, they can only use some high-cost and unstable organics (such as triethanolamine) as the hole sacrificial agent. Instead, HI acts as a carbon-free and safe hydrogen carrier, which attracts more and more attention (Energ. Environ. Sci. 2015, 8, 1484 and Energ. Environ. Sci. 2009, 2, 491). However, those reported perovskite photocatalysts (Nature Energy, 2016, 2, 16185 and Adv. Mater. 2018, 30, 1704342) cannot stably work in HI solution. Considering those issues, we first demonstrate a HI-stable lead-free anti-dissolution environmentally friendly perovskite Cs₂SnI₆ photocatalyst with atomically dispersed Pt-I₃ species, which achieves a stable and high activity (430 μmol h⁻¹ g⁻¹) in a 24-hour HI-to-hydrogen test.

(3) Fundamental photocatalysis. By combining charge-carrier dynamics with theory calculations, this work reveals an origin for the dramatically boosted photocatalytic performance on PtSA/Cs₂SnI₆, which results from both unique coordination structure and electronic property of Pt-I₃ sites, and strong metal-support interaction effect. This work opens a new pathway for stimulating more research on perovskite composite materials for efficient hydrogen production.

Moreover, we have performed all the experiments and theoretical calculations suggested by you, further addressed the comments point-by-point and made the corresponding changes accordingly in the revised manuscript.

Q1: The computational details for the calculated results are frankly insufficient. The choice of adopted functional (which by the way does not include *Van der Waals* interactions that might play a role) is not motivated, convergence tests for the employed slab and the size of the vacuum layer are not reported.

R1: Thanks for your valuable comment. In the initial calculations, we have considered the *Van der Waals* interactions, slab thickness, the vacuum thickness and supercell area in the convergence tests. The results demonstrate the accuracy of present settings. The addition of *Van der Waals* correction shows a limited change in the hydrogen adsorption calculation on all-inorganic Cs₂SnI₆ structure. Besides, the larger supercells do not produce obvious influence on the hydrogen adsorption calculation. Those parameters and results have added into the revised supporting information (Please see **Line 11-13 of Page 11** and **Supplementary Table 3**).

Supplementary Table 3. Convergence tests with different slab thickness, vacuum thickness and supercell area.

Slab thickness (CsI-Sn-CsI layer)	Slab vacuum thickness (Å)	Supercell area	Pt size	Van der Waals functional correction ^a	ΔG (eV)
3	16	2 × 2	Pt ₁	No	-0.11
3	16	1 × 1	Pt ₁	No	-0.50
3	16	3 × 3	Pt ₁	No	-0.13
3	12	2 × 2	Pt ₁	No	-0.22
3	20	2 × 2	Pt ₁	No	-0.14
2	16	2 × 2	Pt ₁	No	-0.24
4	16	2 × 2	Pt ₁	No	-0.10
3	16	2 × 2	Pt ₆	No	-0.91

3	16	2×2	Pt ₃₁	No	-0.85
3	16	2×2	Pt ₁	Yes	-0.15
3	16	2×2	Pt ₆	Yes	-0.96

^aDFT-D2 method was used in Van der Waals functional correction.

Q2: Similarly, results are not clear. How do the authors simulate “before and after photexcitation” conditions in their calculations? Does calculation include extra electrons/holes? Then would a GGA treatment be sufficient to describe unpaired charges?

R2: Thanks for your valuable comments. According to the previous report (*Nat Mater* 2016, **15**, 1107-1112), an extra electron with compensating uniform background charge can be used to simulate the photogenerated electron. However, the extra electron with background charge probably leads to the undesirable effect on the electron transfer between Pt cocatalyst and photocatalyst. Hence, in order to avoid the presence of background charge in the region between adjacent slabs, a donor hydrogen atom was inserted into the bulk structure to calculate the charge density difference. The obtained charge density difference mapping also shows the localized distribution of electron in the PtSA region. Similarly, the electron is distributed on the whole PtNP. This has been added into in the revised manuscript (please see **Line 17-27 of Page 8**, **Line 17-24 of Page 11** and **Supplementary Fig. 21**).

Supplementary Fig. 21. The charge density difference maps between PtSA/PtNP and Cs₂SnI₆: (a) PtNP/Cs₂SnI₆ and (b) PtSA/Cs₂SnI₆. The isosurface of charge density is $0.001 \text{ e } \text{\AA}^{-3}$. The *insets* stand for the top view. The yellow region represents the additional electron distribution. An excess donor hydrogen atom was added into the models.

Q3: (a) The authors report charge density differences for the pristine material and PtSA but they do not include any comparison with PtNP. Furthermore, the localization of the charge in the figure is not clear. (b) Are the author claiming that Cs₂SnI₆ feature a delocalized electronic state while, in presence of the additive, they observe a localized state in the band gap of the material?

R3: Thanks for your valuable comments.

(a) Fig. 5a and 5b show the charge density differences between PtNP (PtSA) and Cs₂SnI₆, respectively. The electron is observed to be distributed on the whole PtNP and neighboring I sites, which undesirably decreases the electron density *per* Pt atom in PtNP. Instead, the electron in PtSA/Cs₂SnI₆ is only located between the PtSA and neighboring three I atoms. Thus, the electron density *per* Pt atom in PtNP is further lower than that of PtSA, which is considered to lead to the lower HER activity of PtNP. This explanation has been added into the revised manuscript (Please see **Line 18-24 of Page 8**).

(b) In the calculation of PDOS of PtNP-Cs₂SnI₆ and PtSA-Cs₂SnI₆ models, no extra electron was added. The computational details of PDOS have been added into the revised manuscript (please see the **Line 15-17 of Page 11**). The most localized Pt 5d states of PtSA and PtNP are observed in the band gap region of materials, which can accept the photogenerated electrons from the conduction band of materials.

Q4: I think that the claim that PDOS explains the difference between PtSA and PtNP is quite weak. I cannot see a dramatic difference among the two.

R4: Thanks for your valuable comments. A dashed rectangle was added into **Fig. 5c** to label the difference between the localized Pt 5d states of PtSA and PtNP. As shown in the dashed rectangle region of **Fig. 5c**, the partial Pt 5d states of PtNP is above the Fermi level, which indicates its electron-deficient property. Instead, the most Pt 5d states of PtSA is below the Fermi level, indicates its electron-rich property. The higher electron density of PtSA also implies the stronger electron-captured ability, which contributes to the higher hydrogen production activity of PtSA. This discussion has been added into the revised manuscript (please see **Line 1-5 of Page 9 and Fig. 5c**).

Q5: How do the authors calculate the energy diagrams in Fig. 5d?

R5: Thanks for your valuable comment. The reported standard hydrogen electrode (SHE) model (J Phys Chem B 2004,108, 17886-17892.) was adopted in the calculations of Gibbs free energy changes (ΔG) in hydrogen adsorption. The chemical potential of a proton-electron pair, $\mu(\text{H}^+) + \mu(\text{e}^-)$, is equal to the half of the chemical potential of one gaseous hydrogen, $1/2\mu(\text{H}_2)$, at $U = 0 \text{ V vs SHE}$ at $\text{pH} = 0$. This has been added into the calculation details (please see **Line 3-8 of Page 11**).

Q6: The author state that “instability of the organic component” is at the root of the limited application of perovskites in photocatalysis. However, recent reports show that organic-inorganic perovskite can be water-stable.

R6: Thanks for your valuable comments. Though there are few reports (Nature Energy, 2016, 2, 16185 and Adv. Mater. 2018, 30, 1704342) on water-stable perovskites, they cannot stably work in HI solution. Here, we firstly report a HI-stable metal single atoms-modified lead-free perovskite photocatalyst for converting HI into hydrogen.

Q7: The authors should comment on the trends observed for Pt loading? Why, after a maximum, the photocatalytic activity decreases?

R7: Thanks for your valuable comments. When the content of Pt loading on Cs₂SnI₆ surface was higher or lower than 0.12wt%, the photocatalytic activities was decreased. This is considered from the reason that the excessive Pt species reduces the light absorption of Cs₂SnI₆ due to the shading effect. In contrast, the insufficient Pt species cannot provide the rich H₂-releasing active sites. We have supplemented the explanation of catalytic activity trend with the increase of Pt loading in the revised manuscript. (Please see **Line 3-8 of Page 7**).

Minor points:

Q1: The authors should specify that the band alignment achieved *via* XPS is an approximation, since it does

not include effects of the water-perovskite interface in the band alignment.

R1: Thanks for your valuable comments. We have specified that the band alignment is just an approximation since the complicated electrolyte-perovskite interface effects were not considered (Please see **Line 11-12 of Page 5**).

Q2: “The most 5d states of PtSA are below the Fermi level, indicating its strong electron-captured ability” this sentence is not clear and should be rephrased.

R2: Thanks for your valuable comment. The electronic states below Fermi level are filled with electrons. According to **Fig. 5c**, the most 5d states of PtSA are below the Fermi level, indicating its electron-rich state. In contrast, the partial 5d states of PtNP are above the Fermi level, indicating its electron-deficient property. This means that the PtSA species owns a stronger ability for capturing electrons from the Cs₂SnI₆, which contributes to the higher hydrogen production activity of PtSA/Cs₂SnI₆ in the above photocatalytic experiments. This explanation has been added into the revised manuscript (Please see **Line 1-5 of Page 9**).

Overall, on the basis of my comments, I cannot suggest publication of the manuscript in the present form which needs to be profoundly revised in order to explain the results.

REVIEWERS' COMMENTS

Reviewer #1 (Remarks to the Author):

The authors have sufficiently addressed all my questions/comments. The revised manuscript can be accepted for publication in Nature Communications.

Reviewer #3 (Remarks to the Author):

The authors have substantially improved the quality of the manuscript but a few responses need to be refined. Therefore, I recommend the publication of the manuscript upon minor corrections.

1) While an extra hydrogen is added to ensure charge neutrality, the calculation still involves a localized electronic state. Is the GGA treatment enough accurate to describe the involved energetics?

2) The authors reply on the comment on the instability of the organic component that "Though there are few reports (Nature Energy, 2016, 2, 16185 and Adv. Mater. 2018, 30, 1704342) on water-stable perovskites, they cannot stably work in HI solution." Therefore, they should specify in the manuscript that they refer to stability of perovskites in hydrohalic acids.

3) In R4, the authors highlights again qualitatively the difference in the PDOS. Could they provide a more quantitative (e.g. by means of integration of the DOS) comparison?

To Reviewer 3:

The authors have substantially improved the quality of the manuscript but a few responses need to be refined. Therefore, I recommend the publication of the manuscript upon minor corrections.

R: Thanks for your great efforts in reviewing our manuscript. We specially appreciate your valuable comments and suggestions.

Q1: While an extra hydrogen is added to ensure charge neutrality, the calculation still involves a localized electronic state. Is the GGA treatment enough accurate to describe the involved energetics?

R1: Thanks for your valuable comment. The GGA has been successfully used to simulate the localized electronic states in the photocatalytic reaction and even track the transfer of photogenerated electrons (Nature Materials 2016, 15, 1107). We believe that the GGA treatment can satisfy the present calculation requirement.

Q2: The authors reply on the comment on the instability of the organic component that "Though there are few reports (Nature Energy, 2016, 2, 16185 and Adv. Mater. 2018, 30, 1704342) on water-stable perovskites, they cannot stably work in HI solution." Therefore, they should specify in the manuscript that they refer to stability of perovskites in hydrohalic acids.

R2: Thanks for your valuable comment. We have added the corresponding discussion in the revised manuscript. Please see the **Line 3-8 of Page 3**.

Q3: In R4, the authors highlight again qualitatively the difference in the PDOS. Could they provide a more quantitative (*e.g.* by means of integration of the DOS) comparison?

R3: Thanks for your valuable comment. The integrated PDOS area of Pt 5d states above the Fermi level in PtNP and PtSA is calculated to be 1.19 and 0.71, respectively, which quantifies the different electron-saturation levels of PtNP and PtSA. This information has been added into the revised manuscript. Please see the **Line 3-7 of Page 9**.